# Construction of Hierarchical 2D-3D@3D Zn_3_In_2_S_6_@CdS Photocatalyst for Boosting Degradation of an Azo Dye

**DOI:** 10.3390/molecules30071409

**Published:** 2025-03-21

**Authors:** Andreas Katsamitros, Nikolaos Karamoschos, Labrini Sygellou, Konstantinos S. Andrikopoulos, Dimitrios Tasis

**Affiliations:** 1Department of Chemistry, University of Ioannina, 45110 Ioannina, Greece; andrewkatsamitros@gmail.com (A.K.); k4520fd@yahoo.gr (N.K.); 2Foundation of Research and Technology Hellas, Institute of Chemical Engineering Sciences (ICE-HT), P.O. Box 1414, Rio, 26504 Patras, Greece; sygellou@iceht.forth.gr (L.S.); kandriko@upatras.gr (K.S.A.); 3Department of Physics, University of Patras, 26504 Patras, Greece; 4Institute of Materials Science and Computing, University Research Center of Ioannina (URCI), 45110 Ioannina, Greece

**Keywords:** cadmium sulfide, zinc indium sulfide, photocatalysis, characterization, dye degradation

## Abstract

Herein, flower-like Zn_3_In_2_S_6_ (ZIS_3_) crystallites were grown onto acorn leaf-like CdS assemblies via a two-step hydrothermal approach. Under visible light irradiation, the Zn_3_In_2_S_6_-enriched heterostructures demonstrated an enhanced azo-dye degradation rate, with the majority of the organic analyte (Orange G) being degraded within 60 min. In contrast, the CdS-enriched hybrids showed poor photocatalytic performance. The optimized hybrid containing a nominal CdS content of 4 wt% was characterized by various physicochemical techniques, such as XRD, SEM, XPS and Raman. XPS analysis showed that the electron density around the Zn and In sites in Zn_3_In_2_S_6_ was slightly increased, implying a certain charge migration pattern. Complementary information from scavenging experiments suggested that hydroxy radicals were not the exclusive transient responsible for oxidative degradation of the organic azo-dye. This research provides new information about the development of metal chalcogenide-based heterostructures for efficient photocatalytic organic pollutant degradation.

## 1. Introduction

Over the past few decades, the environment has been facing a serious pollution issue due to continuous industrial growth and ever-increasing human consumption habits. Increased concentrations of pharmaceutical substances have been detected in aquatic environments such as lakes and rivers. Among others, the wastewater dumping from industrial facilities poses a serious threat of contamination of wetlands by hazardous substances. Textile industries utilize synthetic aromatic dyes for their processes, with most of the organic residues eventually ending up in the environment as wastewater [1]. The increasing deposition of such toxic and/or mutagenic substances may give rise to deterioration of the quality of life for society. To this end, a variety of processes have been developed for wastewater treatment, such as adsorption, flocculation and advanced oxidation processes (AOPs). The latter are applied primarily to the decomposition of either organic or inorganic contaminants through the in-situ generation of oxidative transient species [2]. Such transient intermediates belong to the general group of reactive oxygen species (ROS), with representative examples being the hydroxy and sulfate radicals. In the family of AOPs, various similar-philosophy strategies have been developed in the recent decades, with the most widely used being the Fenton (Fe^2+^/H_2_O_2_) process, electrocatalysis, ozonolysis and photocatalysis, respectively. In any case, the formed ROS catalyze the oxidative decomposition of pollutants, with mineralization being the ideal scenario of the whole process.

Photocatalysis is an environmentally-friendly choice for achieving a large set of chemical transformations, which are strongly correlated with energy conversion schemes. Regarding heterogeneous photocatalysis, semiconducting catalysts are mostly dispersed in aqueous environments. Irradiation under a proper light energy source may give rise to a sequence of physicochemical processes, such as exciton dissociation, charge carrier migration and the formation of ROS by redox half-reactions [3]. In the early days of photocatalysis, some fifty years ago, various metal oxides, such as ZnO and TiO_2_, were used extensively as photocatalysts due to the fact that these oxides are abundant, low cost and stable [4]. Despite these advantages, such oxide semiconductors demonstrate a lack of absorption in the visible wavelengths. For instance, TiO_2_ has a large band gap of about 3.2 eV, which corresponds to an absorption edge at ~390nm, approximately. Therefore, the abovementioned semiconductor selectively adsorbs in the UV region. Thus, the related research efforts gradually shifted to alternative semiconducting materials, like CdS and other sulfides [5]. CdS has gained wide recognition owing to its suitable band structures for the various redox half-reactions of photocatalysis and the moderate band gap of ~2.4 eV, extending the absorption edge to the visible region (~517 nm). However, in its pristine state, the charge carrier recombination phenomena need to be controlled and totally altered. Furthermore, CdS displays inherent severe photocorrosion, which altogether inhibits its photoactivity. Over the years, various morphologies of the abovementioned metal sulfide material, coupled with other nanostructured semiconductors, have been implemented as heterostructured photocatalysts in various photocatalytic applications [6]. It has been suggested that the additional semiconducting component acts as a protecting layer toward the inhibition of corrosion process.

With this continuous search for novel materials, a new family of ternary mixed metal sulfides emerged as potential optically active systems, denoted as Zn_x_In_2_S_3+x_ (ZIS_x_, x = 1–5) [7]. Their reported optical band gaps were estimated to range between 2.46 and 2.86 eV. Like CdS, ZIS_x_ exhibit fast recombination of the photogenerated species, but unlike CdS, ZIS_x_ semiconductors have low toxicity and high photochemical stability, owing to their unique layered structure protecting the sulfur sites from photooxidation while at the same time aiding the generation of sulfur vacancies on the crystal lattice. Of this family, the ZnIn_2_S_4_ semiconductor has been the most studied system [8]. However, homologues of these series, such as Zn_3_In_2_S_6_ (ZIS_3_), have been alternatively proposed as photoactive catalytic systems [9,10,11,12,13,14]. Concerning the tuning of optical properties and the enhancement of photocatalytic activity of the parent semiconductor, researchers have developed a few Zn_3_In_2_S_6_-based hybrid materials with other cocatalysts. In this context, the goal was to facilitate the electron/hole dissociation and the subsequent charge carrier migration to the redox catalytic sites. In recent years, there have been some interesting works about the participation of Zn_3_In_2_S_6_-based systems in photocatalytic applications. These include hydrogen evolution [15,16,17,18,19,20], carbon dioxide reduction [21], Cr(VI) reduction [22], H_2_O_2_ evolution [23], transformation of organics [24] and organic pollutant degradation [25,26,27,28,29]. To our knowledge, the integration of Zn_3_In_2_S_6_ with cadmium chalcogenide nanostructures for photocatalytic applications has not been studied in detail. In the recent work of Luan et al. [20], the authors compounded In-doped CdSe with Zn_3_In_2_S_6_ microspheres. The resulting hybrid demonstrated enhanced photocatalytic activity for hydrogen evolution.

Herein, following our previous work on ZnIn_2_S_4_ [30], we focused on the development of novel ZIS_3_-based hybrids by using CdS as a cocatalyst [31]. Specifically, neat ZIS_3_ as well as ZIS_3_/CdS hybrids were synthesized by a simple hydrothermal method. All the samples were characterized by various physicochemical techniques, such as X-ray diffraction (XRD), Scanning Electron Microscopy (SEM), X-ray Photoelectron Spectroscopy (XPS), Raman and Diffuse Reflectance Spectroscopy (DRS). Their performance was assessed in the field of photocatalytic organic dye degradation under visible light irradiation. The dye was Orange G, an azo organic compound, and the quantitative analysis took place in aqueous solution by monitoring the degradation kinetics through the absorption measurements. The correlation between chemical speciation and photocatalytic performance provides valuable insight for developing functional photocatalytic systems.

## 2. Results and Discussion

### 2.1. Physicochemical Characterization

Neat semiconductors as well as hybrid materials were characterized using an X-ray diffraction (XRD) technique. Regarding the neat Zn_3_In_2_S_6_ (ZIS_3_) sample (Figure 1A), the characteristic peaks were located at 22.9°, 26.8°, 28.4°, 47.2°, 56.1° and 76.3° corresponding to (005), (100), (102), (110), (203) and (213) crystallographic planes, respectively [32]. The crystallinity profile of neat CdS (Figure 1B) resembled the one shown in our previous study [30]. Specifically, the typical diffractions of a hexagonal phase were observed at 24.7°, 26.4°, 28.1°, 36.6°, 43.6°, 47.8°, 50.8°, 51.8°, 52.8°, 54.5°, 58.3°, 66.7°, 69.3°, 70.9°, 72.4° and 75.5°, corresponding to (100), (002), (101), (102), (110), (103), (200), (112), (201), (004), (202), (203), (210), (211), (114) and (105) crystallographic planes, respectively. Hybrids highly enriched in a specific semiconducting species mostly demonstrate the corresponding diffraction peaks of the excess component. Representative examples are demonstrated in Figure 1C,D, in which the XRD profiles of “ZIS_3_ 96 wt%” and “ZIS_3_ 84 wt%” are illustrated, respectively. Certain minor peaks may be ascribed to the presence of low-weight fraction CdS components, which were indicated by dotted signs. The main peaks clearly observed have maxima at 24.9°, 26.6°, 36.8°, 43.8°, 51.0°, 52.0°, 53.0°, 58.4°, 66.9°, 69.5°, 71.2°, 72.8° and 75.7°, respectively. It is notable that the weak intensity peaks were most clearly observed in the “ZIS_3_ 84 wt%” sample (Figure 1D), which supports the comparatively higher CdS fraction in the aforementioned hybrid. Samples containing a theoretical ZIS_3_ weight fraction of less than 50% demonstrated mainly the diffraction peaks of a CdS component.

The morphology of the synthesized nanostructured materials was assessed by Scanning Electron Microscopy (SEM) imaging. In Figure 2A, spherical flower-like morphologies were observed, grown through the assembly of thin leaf crystallites [33]. The size distribution of the crystallites ranged between approximately 1 and 5 μm. In contrast, the morphology of neat CdS crystallites resembled an acorn leaf structure, with a uniform size and shape (Figure 2B) [34]. Hydrothermal growth of ZIS_3_ crystallites in the presence of a CdS component gave rise to a partial phase-separated heterostructure in certain domains, a pattern which was more noticeable with lower ZIS_3_ mass loadings. Concerning the “ZIS_3_ 96 wt%” sample (Figure 2C), flower-like crystallites were grown in the vicinity of CdS assemblies. Elemental mapping by a combination of SEM/EDX imaging (Appendix A) demonstrated that zinc, indium and sulfur elements were homogeneously dispersed in the recorded heterostructures. In comparison, a cadmium component presents a somewhat lower distribution homogeneity, showing some partial phase-separation phenomena. This was more obvious in the hybrid sample, containing a theoretical ZIS_3_ fraction of 50 wt% (Figure 2D).

The chemical speciation and the surface analysis of the studied samples was assessed using XPS techniques. The survey scans of neat ZIS_3_ and “ZIS_3_ 96 wt%” samples showed the involved elements, namely zinc, indium, sulfur and cadmium (Appendix A). The latter elemental component was recorded only in the case of hybrid structures. It is noted that adventitious carbon was recorded in the survey scans, which could be attributed to exposure of the samples to air. Thus, the carbon component will not be considered part of the percent atomic concentration of elements. From the peak areas of the involved elemental components, the percent relative atomic concentration of each element was derived. Concerning the neat ZIS_3_, the estimated relative atomic concentrations of zinc, indium and sulfur were 31.0% (27.3%), 19.6% (18.2%) and 49.4% (54.5%), respectively. In parentheses, the nominal percent relative atomic concentrations for the parent lattice were given. In the hybrid material containing a theoretical ZIS_3_ fraction of 96 wt%, the corresponding atomic concentrations were estimated to be 31.7%, 19.9% and 47.5%, respectively. In addition, a 0.9% concentration of cadmium was estimated. The aforementioned values were in good agreement with the nominal ones estimated by the stoichiometry of the elements.

Figure 3A,B shows the Zn 2p XPS core level peaks as well as the Zn L_3_M_45_M_45_ X-ray excited Auger electron (XAES) peaks for the studied samples. The binding energy of the Zn 2p_3/2_ peak was centered at 1022.1 eV for neat ZIS_3_, whereas the corresponding peak of the “ZIS_3_ 96 wt%” hybrid was centered at 1021.9 eV (Figure 3A) [21]. The observation of a slight peak shift (by about 0.2 eV) toward lower binding energy values suggested an increase of electron density in the vicinity of Zn within the heterostructure. Concerning the XAES data (Figure 3B), the Zn L_3_M_45_M_45_ Auger peak was centered at 989.2 eV. Accurate information about the chemical state of elements within a formula may be extracted by the estimation of a modified Auger parameter (sum of the abovementioned peak maxima values). The value of the Auger parameter was estimated to be 2011.1 eV, which was strongly indicative of Zn^2+^ ions in the lattice of zinc indium sulfide [30].

An analogous slight peak shift of about 0.2 eV was observed for In 3d (Figure 3C). The binding energy of the In 3d_5/2_ peak was centered at 445.1 eV for neat ZIS_3_, whereas the corresponding peak of the “ZIS_3_ 96 wt%” hybrid was centered at 444.9eV. The observed binding energy values of the In 3d spectra correspond to the In(III) oxidation state. Similarly, the S 2p spectra demonstrated a doublet with the peak maxima shifted by about 0.2eV (Figure 3D and Appendix A). In total, the observation of a slight peak shift in the Zn 2p, In 3d and S 2p spectra implied that electron transfer took place from CdS to Zn_3_In_2_S_6_ [35]. The recording of a cadmium (Cd 3d) component in the hybrid sample (Appendix A) demonstrated a weak intensity due to the low atomic ratio of the cadmium element. The Cd 3d_5/2_ peak was centered at 405.3 eV, which strongly supported the existence of divalent cadmium species within the CdS lattice [30].

The vibrational modes of either parent of the heterostructured samples were assessed by Raman spectroscopy. The Raman spectrum of a neat Zn_3_In_2_S_6_ sample is given in Figure 4A. Certain vibrational modes were located in the spectral range of a ~200–420 cm^−1^ window, within which several peaks were resolved at approximately 250, 304, 345 and 375 cm^−1^, comprising a broad spectral feature. The spectrum is in perfect analogy with the one of previous works [36,37]. The band at ~250 cm^−1^ was attributed to the A_1g_ mode of the crystal. The remaining bands above 250 cm^−1^ were ascribed to a superposition of the vibrations of ZnS and In_2_S_3_ species of amorphous-like structures, characterized by defects in the octahedral and tetrahedral cation sublattice.

Concerning the “ZIS_3_ 96 wt%” heterostructure, optical observation of the grains was performed using an optical microscope fitted with high magnification objectives (50× or 100×). White/pale yellow-colored regions were abundant in the solid sample, whereas orange-colored domains at the size range of a very few micrometers were homogeneously distributed in far lower abundance. Selected Raman spectra were acquired from various spots of the powder sample (Figure 4B). The Raman spectrum of neat Zn_3_In_2_S_6_ was also included for comparison. The Raman spectra of isolated orange-colored grains exhibited very strong resonance Raman bands at 300 cm^−1^ and 600 cm^−1^, which are typical of the CdS first order vibrational mode and its overtone. Raman spectra obtained from the abundant white/pale yellow-colored regions exhibited several vibrational modes, which may be assigned to neat Zn_3_In_2_S_6_ (indicated by arrows in the graph), neat CdS as well as sulfur S8 rings (155, 219, 473 cm^−1^ peaks) and possibly sulfur chains (shoulder at ~460 cm^−1^) interconnecting Zn/In cations [30,38]. Despite the low mass fraction of the CdS component, the abovementioned doublet was recorded in the white/pale yellow-colored domains of the powdered sample. It is noted that the elemental sulfur adducts demonstrate enhanced intensity due to their strong scattering properties and not due to their appreciable mass fraction. The existence of trace amounts of elemental sulfur adducts was supported by the fact that XPS analysis has not recorded such species.

### 2.2. Optical Properties and Photocatalytic Performance

Diffuse reflectance spectroscopy (DRS) was used in order to assess the optical properties of the samples studied. In Figure 5A, the reflectance spectrum of neat Zn_3_In_2_S_6_ was illustrated, acquiring a steep absorption edge in the window ranging from approximately 400 to 500 nm. The indirect transition band gap of the neat semiconductor was calculated through the tangent line within the Kubelka–Munk function plot against energy (Figure 5B) [39]. The band gap value was estimated to be 2.74 eV, which corresponds to a visible photon absorbance of about 452 nm energy input. The optical properties of neat CdS revealed a band gap of 2.23 eV [30]. Concerning the “ZIS_3_ 96 wt%” sample, the corresponding DRS and band gap-related data are illustrated in Figure 5C,D. It was apparent that the band gap of the heterostructure was slightly decreased by about 0.08 eV, giving rise to a band excitation energy of about 466 nm.

The photocatalytic performance of either pristine or heterostructured materials was assessed by following the photodegradation kinetics of an organic azo-dye, namely Orange G. As described in the Experimental Section, we followed the dye degradation kinetics by carrying out UV–Vis absorption spectroscopy measurements. At certain time intervals, sampling was carried out and monitoring of the dye absorption at λ_max_ = 480 nm took place. In Figure 6A, the decay profiles of the neat semiconductors and two of the most efficient heterostructures are shown. The corresponding curves of the remaining hybrid materials were demonstrated in Appendix A. It was found that ZIS_3_-enriched hybrids demonstrated enhanced photocatalytic performance for dye degradation when compared with the corresponding CdS-enriched ones. This is in strong accordance with the findings of our previous work [30]. Optimized photocatalytic performance could be achieved by the fabrication of Zn_3_In_2_S_6_@CdS hybrids containing a 4 wt% theoretical content of CdS component. After 60 min of irradiation time, the large majority of dye analyte (95.5% of the starting concentration) was degraded. For comparison, the photocatalytic performance of the studied samples (at C_dye_ = C_0,dye_/2) is given in a decreasing order of activity: ZIS_3_ 96 wt% > ZIS_3_ 99 wt% > neat ZIS_3_ > ZIS_3_ 84 wt% > ZIS_3_ 50 wt% > ZIS_3_ 4 wt% > ZIS_3_ 16 wt% > ZIS_3_ 1 wt% > neat CdS.

Treatment of the optimized “ZIS_3_ 96 wt%” hybrid under consecutive photocatalytic cycles has demonstrated the potential of the sample to be reused after the recovery process. In Figure 6B, the dye decay profiles of four (4) consecutive cycles of recovered photocatalyst are shown. A remarkable performance maintenance was observed, with a slight decrease of dye degradation ability at 120 min of irradiation. Specifically, in the third cycle, some slight performance decrease was shown. However, for shorter irradiation times (at t = 60 min), it is noted that the amount of degraded dye was somewhat diminished even in the second cycle. In general, the observed performance alteration could be ascribed to possible physical adsorption phenomena onto the catalytic sites of the heterostructure. The investigation of the potential mechanism leading to the ROS-mediated dye degradation has taken place through the elaboration of photocatalytic experiments in the presence of scavenging species. A partial inhibition of dye degradation was observed by using both isopropanol (hydroxy radical scavenger) and sodium azide. The latter scavenger is responsible for the trapping of both hydroxy radicals and singlet oxygen [40]. Similar decay profiles were recorded for both scavenging species. We thus assumed that the transients responsible for the oxidative degradation of Orange G dye were mainly hydroxy radicals, with singlet oxygen playing a minor role in the whole process.

Taking into account the complementary information received by XPS, DRS and scavenging experiments, one may draw a potential mechanistic scheme of the photocatalytic degradation mechanism. After the excitation step, electrons migrate from the CdS conduction band to the corresponding one of Zn_3_In_2_S_6_. On the other hand, holes migrate from the Zn_3_In_2_S_6_ valence band to the corresponding one of the CdS. Thus, electrons accumulate in the Zn_3_In_2_S_6_ conduction band, whereas holes accumulate in the CdS valence band. The latter catalytic site may participate in both oxidative degradation of organic pollutants as well as the hole-mediated oxidation of water toward hydroxy radical formation. The transient intermediate may subsequently oxidize the azo-dye. However, the observed partial inhibition of degradation in the presence of hydroxy radical scavengers implies that the latter transient intermediate should not be the exclusive factor for theoxidative degradation of dye. The organic pollutant may be further oxidized by transients generated through the oxygen reduction path. The accumulated electrons in the Zn_3_In_2_S_6_ conduction band may participate in the reductive evolution of ROS such as superoxide radicals, H_2_O_2_ and hydroxy radicals.

## 3. Materials and Methods

### 3.1. Precursor Chemicals

The following reagents and solvents were purchased from Sigma Aldrich (St. Louis, MO, USA) and used as received. These are indium(III) nitrate hydrate In(NO_3_)_3_∙6H_2_O, zinc chloride ZnCl_2_, thioacetamide CH_3_CSNH_2_ (TAA), cadmium chloride CdCl_2_, thiourea NH_2_CSNH_2_, Orange G dye (OG) and methanol.

### 3.2. Zn_3_In_2_S_6_ (ZIS_3_) Synthesis

For the synthesis of Zn_3_In_2_S_6_ (ZIS_3_), a simple hydrothermal method was implemented wherein 636 mg In(NO_3_)3∙6H_2_O (2 mmol), 408 mg ZnCl_2_ (3 mmol) and an excess of TAA (900 mg, 12 mmol) were dissolved in 12 mL of deionized water with the aid of a magnetic stirring mantle for 5 min, followed by ultrasonic treatment for 2 min [25]. Subsequently, the aqueous solution was added into a Teflon-lined stainless-steel autoclave, sealed and heated in the oven for 12 h at 180 °C. After cooling overnight, the resulting suspension was centrifuged at 6000 rpm and washed with deionized water and methanol, two times each. Lastly, the solid precipitate was dried overnight at 90 °C and a yellow powder was obtained.

### 3.3. CdS Synthesis

For the hydrothermal synthesis of CdS, the protocol of a previous work was adopted and modified [41]. In brief, 242 mg CdCl_2_ (1.3 mmol) and 100 mg thiourea (1.3 mmol) were dissolved in 12 mL of deionized water with the aid of a magnetic stirring mantle for 5 min, followed by ultrasonic treatment for 2 min. Subsequently, the aqueous solution was added into a Teflon-lined stainless-steel autoclave, sealed and heated in the oven for 12 h at 180 °C. After cooling overnight, the resulting suspension was centrifuged at 6000 rpm and washed with deionized water and methanol, two times each. Lastly, the solid precipitate was dried overnight at 90 °C and an orange powder was obtained.

### 3.4. Zn_3_In_2_S_6_@CdS Hybrids Synthesis

For the synthesis of Zn_3_In_2_S_6_@CdS hybrids, a similar protocol as the one used in the case of pristine ZIS_3_ (vide supra) was followed, with the exception of adding appropriate amounts of CdS material with the precursors for ZIS_3_ growth. In the resulting composite materials, the theoretical weight fractions of the ZIS_3_ component were 1 wt%, 4 wt%, 16 wt%, 50 wt%, 84 wt%, 96 wt% and 99 wt%, respectively. For simplification purposes, each sample was denoted as “ZIS_3_ x wt%”, where x represents the nominal weight fraction of the ZIS_3_ component in the hybrid.

### 3.5. Characterization Methods

Powder XRD measurements were recorded using a BRUKER AXS (D8 ADVANCE, Billerica, MA, USA) unit, equipped with a Cu X-ray tube. The morphology of the hybrids was assessed by Scanning Electron Microscopy (SEM) imaging (model JSM-6510LV, JEOL, Tokyo, Japan) (UoI microscopy unit).

For the inelastic scattering spectra, the micro-Raman T-64000 system (Horiba, Paris, France) was used. The 514.5 nm wavelength was selected for the excitation of the samples, which were illuminated by means of a laser beam of ~0.1 mW power appropriately focused by a 50x microscope objective. Backscattered radiation was collected by the same objective and was directed to a single spectrograph equipped with a liquid nitrogen cooled CCD detector after passing an edge filter (Horiba, Paris, France). The utilized configuration enabled the recording of Stokes-side vibrational spectra above ~180 cm^−1^ with a resolution of ~7 cm^−1^.

The UV–Vis diffuse reflectance spectra (DRS) of the fabricated catalyst (powder) were recorded using a Shimadzu 2600 spectrophotometer bearing an IRS-2600 integrating sphere (Kyoto, Japan) in the wavelength of 200–800 nm at room temperature using BaSO_4_ (Nacalai Tesque, extra pure reagent, Kyoto, Japan) as a reference sample.

The surface chemical composition of the nanoparticles was measured by X-ray photoelectron spectroscopy (XPS). Samples in powder form were suspended in acetone by sonication and drop casted on ITO substrate (Indium Tin Oxide-coated glass). The analysis was performed in an ultra-high vacuum chamber equipped with a SPECS Phoibos 100 hemispherical analyzer-1D Delay Line Detector (SPECS Surface Nano Analysis, GmbH-Berlin, Germany) and a dual anode Mg/Al X-ray source (SPECS X-ray source RQ 20/63, Berlin, Germany). The spectra were collected with a Mg Kα X-ray source (1253.6 eV) and the data were processed using Specs Lab Prodigy 4.113.1 software. The XPS spectra were calibrated according to the C 1s reference (284.6 eV). The fitting was done using a Shirley background and convoluted with a mixed Gaussian–Lorentzian profile.

### 3.6. Photocatalytic Activity

In a typical photocatalytic Orange G dye degradation experiment, the solution of Orange G dye was first prepared by diluting an amount of the dye in a volume of deionized water in order to prepare a 0.33 mg/mL Orange G stock solution. Subsequently, 3 mL of the dye solution and 97 mL of deionized water were added in a 250 mL round-bottomed flask and placed into the sonication bath for 5 min. Subsequently, 50 mg of either pristine or hybrid photocatalyst were added into the solution, which was bath-sonicated for another 5 min. The resulting suspension was then transferred into a 250 mL reactor (Lenz, Germany). The reactor was placed in a solar simulator with 2 lamps of 800 W each used as the excitation source, with continuous stirring and a tap water cooling circuit to maintain the temperature around 23 °C. At first, the suspension in the reactor was kept under stirring in the dark for 30 min before irradiation. At specific time intervals (−30, 0, 15, 30, 45, 60, 90 and 120 min), aliquots of 3 mL were taken out of the reactor with a 10 mL syringe, filtered through a 0.22 μm PTFE syringe filter and stored in a glass vial in a dark spot. Lastly, using a spectrophotometer, their absorbance at 480 nm was measured and the relative concertation (C/C_0_) was plotted against time, through the establishment of a calibration curve.

## 4. Conclusions

In summary, an efficient noble metal-free Zn_3_In_2_S_6_@CdS photocatalyst with 2D-3D@3D geometry has been successfully synthesized via the hydrothermal method. XPS analysis showed that the electron density around the Zn and In sites in Zn_3_In_2_S_6_ was slightly increased, implying a certain charge migration pattern from the CdS conduction band to the corresponding one of Zn_3_In_2_S_6_. Complementary information by Raman and EDX mapping strongly suggested the homogeneous distribution of ionic metal species in a few-μm domains. Nevertheless, some phase-separation was observed in hybrids with moderate ZIS_3_ loadings (e.g., 50 wt%). The optimized hybrid photocatalytic reaction with a nominal CdS content of 4 wt% shows the highest azo-dye degradation rate, with most of the analyte being degraded in 60 min. In comparison, CdS-enriched hybrids demonstrated a poor photocatalytic performance. The photocatalytic performance of recovered catalyst was unaltered through up to four cycles. Scavenging experiments suggested that hydroxy radicals were the dominant species for the oxidative degradation of azo-dye. Thus, the significant findings of this study provide new insights into developing highly efficient and noble-metal-free photocatalysts for organic pollutants using an ROS-mediated process.

## Figures and Tables

**Figure 1 molecules-30-01409-f001:**
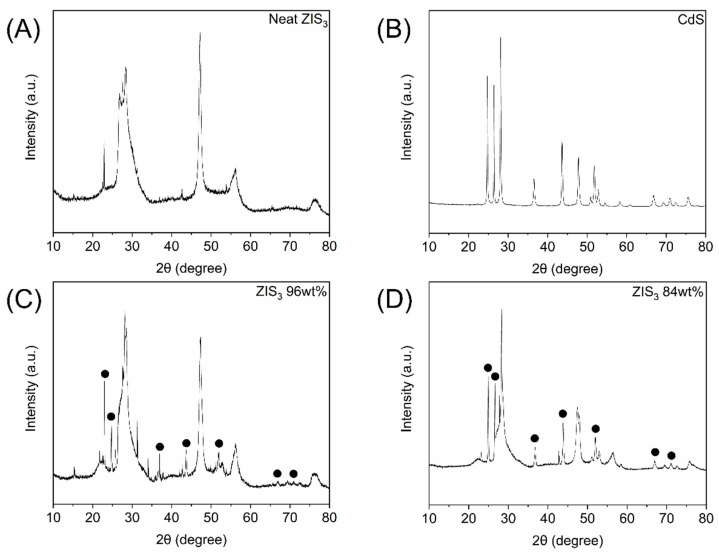
X-ray diffraction profiles of (**A**) neat Zn_3_In_2_S_6_, (**B**) neat CdS, (**C**) “ZIS_3_ 96 wt%” hybrid and (**D**) “ZIS_3_ 84 wt%” hybrid. Peaks ascribed to a CdS component are located by dotted signs (**C**,**D**).

**Figure 2 molecules-30-01409-f002:**
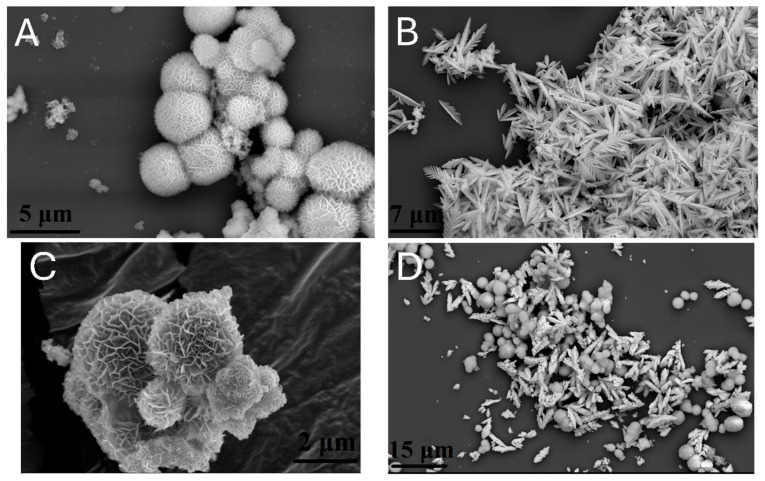
Representative SEM images of (**A**) neat Zn_3_In_2_S_6_, (**B**) neat CdS, (**C**) “ZIS_3_ 96 wt%” and (**D**) “ZIS_3_ 50 wt%” samples.

**Figure 3 molecules-30-01409-f003:**
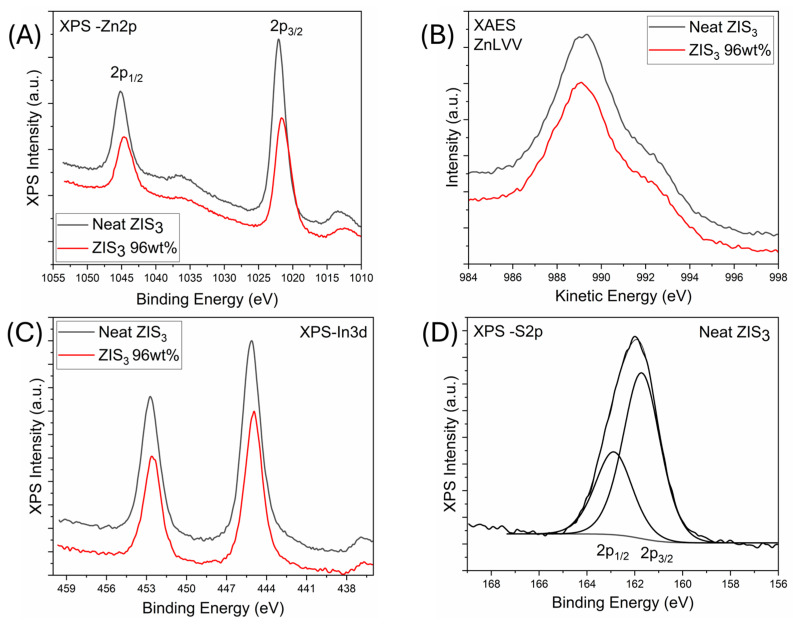
(**A**) Zn2p XPS core level peaks of neat ZIS_3_ and “ZIS_3_ 96 wt%” samples; (**B**) Zn L_3_M_45_M_45_ XAES peaks of neat ZIS_3_ and “ZIS_3_ 96 wt%” samples; (**C**) In3d XPS core level peaks of neat ZIS_3_ and “ZIS_3_ 96 wt%” samples; (**D**) S2p XPS core level peaks of neat ZIS_3_ sample.

**Figure 4 molecules-30-01409-f004:**
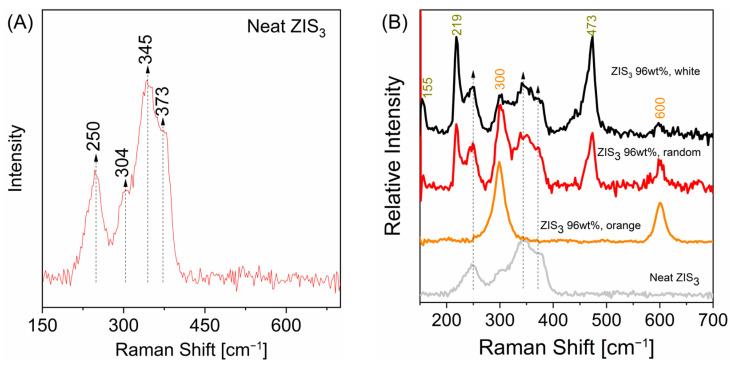
(**A**) Raman spectrum of neat ZIS_3_; (**B**) Mapping of “ZIS_3_ 96 wt%” hybrid in three different regions.

**Figure 5 molecules-30-01409-f005:**
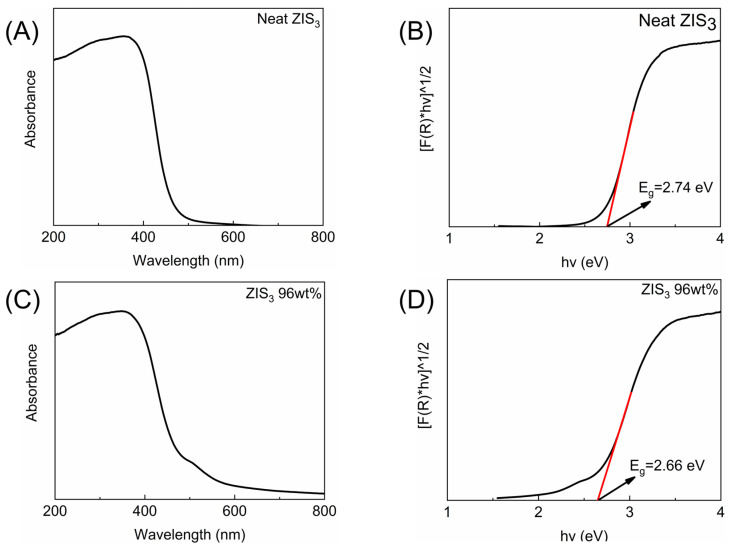
Diffuse reflectance spectra and band gap estimation via the Kubelka–Munk function of neat ZIS_3_ (**A**,**B**) and “ZIS_3_ 96 wt%” hybrid (**C**,**D**).

**Figure 6 molecules-30-01409-f006:**
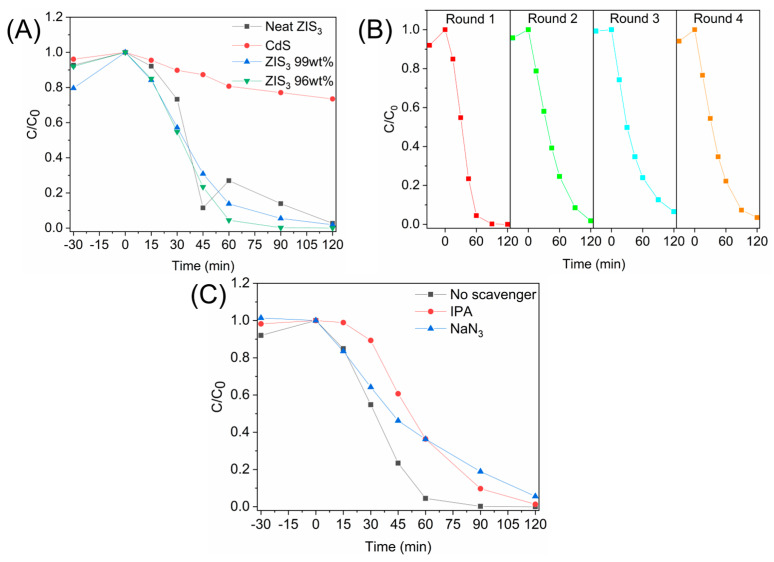
(**A**) Dye degradation profiles for neat CdS, neat ZIS_3_, “ZIS_3_ 99 wt%” and “ZIS_3_ 96 wt%” samples; (**B**) Consecutive cycles of dye photodegradation experiments of the “ZIS_3_ 96 wt%” sample; (**C**) Scavenging experiments for dye degradation experiments in the presence of “ZIS_3_ 96 wt%” sample.

## Data Availability

The data presented in this study are available on request from the corresponding author.

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
