# Peer review of "Construction of Hierarchical 2D-3D@3D Zn3In2S6@CdS Photocatalyst for Boosting Degradation of an Azo Dye"

_molecules, 2025, doi:10.3390/molecules30071409_

Round 1

Reviewer 1 Report

Comments and Suggestions for Authors

Comments

This work prepares composites by growing flower-like Zn3In2S6 crystals onto acorn leaf-like CdS assemblies by hydrothermal method. The experimental results show that under visible light irradiation, the degradation rate of azo dyes with Zn3In2S6-rich hybridized structures is increased, and most of the organic analytes (orange G) are degraded within 60 min. On the contrary, the photocatalytic performance of the CdS-rich heterostructures was poor. However, before considering publication, the following issues need to be addressed.

  1. The title mentions the construction of 0D-3D materials, which include flower-like Zn3In2S6 and acorn leaf-like CdS. The authors need to clarify the specific dimensions of these two materials, are these two materials categorized as 2D materials or some other dimension?

  1. The XPS spectra of some elements in Fig.3 are not well fitted; moreover, according to the description in the paper, all elements in the Zn3In2S6/CdS heterojunction are not displaced, but a slight shift of the spectral peaks of both Zn and In can be seen in the figure.

  1. The design of the degradation experiments in this study is relatively simple and does not adequately consider the effects of multiple variables on catalyst performance. The authors need to evaluate the catalyst performance by the degradation efficiency under different experimental conditions, such as pH, light source intensity, and catalyst dosage.

  1. The photocatalytic degradation mechanism of the catalysts has not been analyzed in depth in this study, and the authors need to analyze the possible electron transfer mechanisms, generation, and role of free radicals in the catalytic degradation process based on experimental data and material characterization.

  1. The title of the material in this paper is “Zn3In2S6/CdS”, but in the conclusion section, the name of the material is stated as “Zn3In2S6@CdS”, and the authors need to clarify the name of the material.

  1. Some recent works focus on the studying of CdS-based photocatalyst are suggested to review for polishing the revised manuscript, such as Chinese Journal of Chemical Engineering 43 (2022) 266–274
Comments on the Quality of English Language

 The English could be improved to more clearly express the research.

Reviewer 2 Report

Comments and Suggestions for Authors

This manuscript entitled “Construction of Hierarchical 0D-3D Zn3In2S6/CdS Photocatalyst for Boosting Degradation of an Azo Dye” deals on the preparation and characterization of a hybrid materials based on flower like zinc-indium sulfide crystallites grown on CdS and their application in the catalytic photodegradation  of azo  dyes. This research is original and into the scope of Molecules. This paper presents some interesting results and it merits its publication but minor and major changes are necessary prior its publication. Some sugegstions are indicated as follows:

1.- The abstract section should be rewritten. The first sentence is very general and must be removed, and part of the rest must be modified to avoid some confusion sentences.

2.- In this sentence “Regarding the neat ZIS3 sample (Figure 1A), the characteristic peaks were located at 22.9o, 26.8o, 28.4o, 47.2o, 56.1o, and 76.3o corresponding to (005), (100), (102), 109 (110), (203) and (213) crystallographic planes, respectively [31]” , please, identify the chemical compound.

3.- This sentence “Concerning the neat ZIS3, the estimated relative atomic concentrations of zinc, indium and sulfur were 31.0% (27.3%), 19.6% (18.2%) and 49.4% (54.5%)” is very confused because the presence of carbon, always present, was not considered.

4.- Please modify this sentence “well as the Zn L3M45M45 X-ray Induced Electron Spectroscopy (XAES) peaks“ . This is an Auger signal but obtained by XPS, not by XAES.

5.- Please, assign the oxidation state of Indium taking into account the observed binding energy values of the In 3d core level spectra.

6.- Please, use the right nomenclature and write indium(III) nitrate instead of indium (III) nitrate.

7.- Please, include the XPS survey spectra as supplementary information. The presence of the C 1s and O 1s signal and the absence of N 1s should be  discussed.  

Round 2

Reviewer 2 Report

Comments and Suggestions for Authors

The revised version of the mansucript has considered all my suggestions and now, I recommend its publication.